# AANAT1 functions in astrocytes to regulate sleep homeostasis

Sejal Davla[1,2,3†], Gregory Artiushin[4†], Yongjun Li[5], Daryan Chitsaz[1,3], Sally Li[1], Amita Sehgal[6], Donald J van Meyel[1,2]*

[1]Centre for Research in Neuroscience, Department of Neurology and Neurosurgery, McGill University, Montreal, Canada; [2]BRaIN Program, Research Institute of the McGill University Health Centre, Montreal, Canada; [3]Integrated Program in Neuroscience, McGill University, Montreal, Canada; [4]Neuroscience Graduate Group, Perelman School of Medicine, University of Pennsylvania, Philadelphia, United States; [5]Biology Graduate Group, University of Pennsylvania, Philadelphia, United States; [6]Howard Hughes Medical Institute, Chronobiology and Sleep Institute, Perelman School of Medicine, University of Pennsylvania, Philadelphia, United States

**Abstract** How the brain controls the need and acquisition of recovery sleep after prolonged wakefulness is an important issue in sleep research. The monoamines serotonin and dopamine are key regulators of sleep in mammals and in *Drosophila*. We found that the enzyme arylalkylamine N-acetyltransferase 1 (AANAT1) is expressed by *Drosophila* astrocytes and specific subsets of neurons in the adult brain. AANAT1 acetylates monoamines and inactivates them, and we found that AANAT1 limited the accumulation of serotonin and dopamine in the brain upon sleep deprivation (SD). Loss of AANAT1 from astrocytes, but not from neurons, caused flies to increase their daytime recovery sleep following overnight SD. Together, these findings demonstrate a crucial role for AANAT1 and astrocytes in the regulation of monoamine bioavailability and homeostatic sleep.

*For correspondence:
don.vanmeyel@mcgill.ca

†These authors contributed equally to this work

## Introduction

Characteristic features of sleep are conserved among species (*Pimentel et al., 2016*), and from humans to insects sleep is influenced by neural circuits involving monoamines such as serotonin and dopamine (*Nall and Sehgal, 2014*). Glial cells are known to take up and metabolize monoamines (*Inazu et al., 2003*; *Pelton et al., 1981*; *Suh and Jackson, 2007*; *Takeda et al., 2002*) and they have been increasingly implicated in mechanisms of baseline and homeostatic sleep regulation in mammals and flies (*Artiushin et al., 2018*; *Briggs et al., 2018*; *Chen et al., 2015*; *Clasadonte et al., 2017*; *Farca Luna et al., 2017*; *Frank, 2013*; *Gerstner et al., 2017*; *Halassa et al., 2007*; *Haydon, 2017*; *Pelluru et al., 2016*; *Seugnet et al., 2011*; *Stahl et al., 2018*; *Vanderheyden et al., 2018*; *Walkowicz et al., 2017*), but it remains unknown whether and how glia might influence monoaminergic control of sleep. Sleep is regulated by circadian clocks and a homeostatic drive to compensate for prolonged wakefulness, and growing evidence suggests that neural mechanisms controlling homeostatic sleep can be discriminated from those controlling baseline sleep (*Artiushin and Sehgal, 2017*; *Donlea et al., 2017*; *Dubowy and Sehgal, 2017*; *Liu et al., 2016*). In *Drosophila*, mutants of arylalkylamine N-acetyltransferase 1 (*AANAT1[lo]*) were reported to have normal baseline amounts of sleep and motor activity, but increased recovery sleep ('rebound') following deprivation (*Shaw et al., 2000*). AANAT1 corresponds to *speck*, a long-known mutation characterized by a darkly pigmented region at the wing hinge (*Spana et al., 2020*). AANAT1 can

**eLife digest** Sleep is essential for our physical and mental health. A lack of sleep can affect our energy and concentration levels and is often linked to chronic illnesses and mood disorders.

Sleep is controlled by an internal clock in our brain that operates on a 24-hour cycle, telling our bodies when we are tired and ready for bed, or fresh and alert to start a new day. In addition, the brain tracks the need for sleep and drives the recovery of sleep after periods of prolonged wakefulness – a process known as sleep-wake homeostasis.

Chemical messengers in the brain such as dopamine and serotonin also play an important part in regulating our sleep drive. While dopamine keeps us awake, serotonin can both prevent us from and help us falling asleep, depending on the part of the brain in which it is released.

Most research has focused on the role of different brain circuits on sleep, but it has been shown that a certain type of brain cell, known as astrocyte, may also be important for sleep regulation. So far, it has been unclear if astrocytes could be involved in regulating the need for recovery sleep after a sleep-deprived night – also known as rebound sleep.

Now, Davla, Artiushin et al. used sleep-deprived fruit flies to investigate this further. The flies were kept awake over 12 hours (from 6pm to 6am), using intermittent physical agitation. The researchers found that astrocytes in the brains of fruit flies express a molecule called AANAT1, which peaked at the beginning of the night, declined as the night went on and recovered by morning. In sleep deprived flies, it inactivated the chemical messengers and so lowered the amount of dopamine and serotonin in the brain.

However, in mutant flies that lacked AANAT1, both dopamine and serotonin levels increased in the brain after sleep deprivation. When AANAT1 was selectively removed from astrocytes only, sleep-deprived flies needed more rebound sleep during the day to make up for lost sleep at night. This shows that both astrocytes and AANAT1 play a crucial role in sleep homeostasis.

Molecules belonging to the AANAT family exist in both flies and humans, and these results could have important implications for the science of sleep. The study of Davla, Artiushin et al. paves the way for understanding the mechanisms of sleep homeostasis that are similar in both organisms, and may in the future, help to identify sleep drugs that target astrocytes and the molecules they express.

acetylate and inactivate monoamines in vitro (*Hintermann et al., 1995*), but the role of AANAT1 in vivo remains poorly understood.

We report here that AANAT1 is expressed in astrocytes and subsets of neurons in the adult *Drosophila* brain, with levels in astrocytes first rising then declining markedly overnight. In sleep deprived *AANAT1* mutant flies, heightened recovery sleep is accompanied by increased serotonin and dopamine levels in the brain. With cell type selective AANAT1 knockdown, we find that AANAT1 functions in astrocytes, but not neurons, to limit the amount of recovery sleep that flies take in response to sleep deprivation (SD). These findings identify a critical role for astrocytes in the regulation of monoamine bioavailability and calibration of the response to sleep need.

## Results and discussion

We generated antiserum to AANAT1 (known previously as Dopamine acetyltransferase (Dat)) and confirmed its specificity with immunohistochemistry (IHC) in the embryonic central nervous system (CNS). AANAT1 immunoreactivity was observed in the cytoplasm of many cells (*Figure 1A*) but was absent in age-matched ventral nerve cords of embryos that were homozygous for a deletion of the entire AANAT1 gene (*Figure 1B*). In adult brains co-immunostained with anti-Bruchpilot (nc82, *Figure 1C*), a presynaptic marker that labels neuropil regions, AANAT1 was present in distinct populations of cells throughout the brain. We found that AANAT1 was expressed in sub-populations of neurons (anti-Elav positive (+), *Figure 1D,D'*) and glia (anti-Repo+, *Figure 1E,E'*) throughout the brain. In glial cells AANAT1 was primarily cytoplasmic, but in neurons AANAT1 often appeared to localize to the nucleus. With the astrocyte-specific Alrm-Gal4 driving expression of a Red Fluorescent Protein (RFP) reporter (Alrm >nuRFP), we confirmed that all AANAT1-positive glial cells in the central brain are astrocytes (*Figure 1F–F''*). Only a subset of astrocytes in the optic lobes that reside between the medulla and lobula did not express AANAT1 (*Figure 1—figure supplement 1A*). With

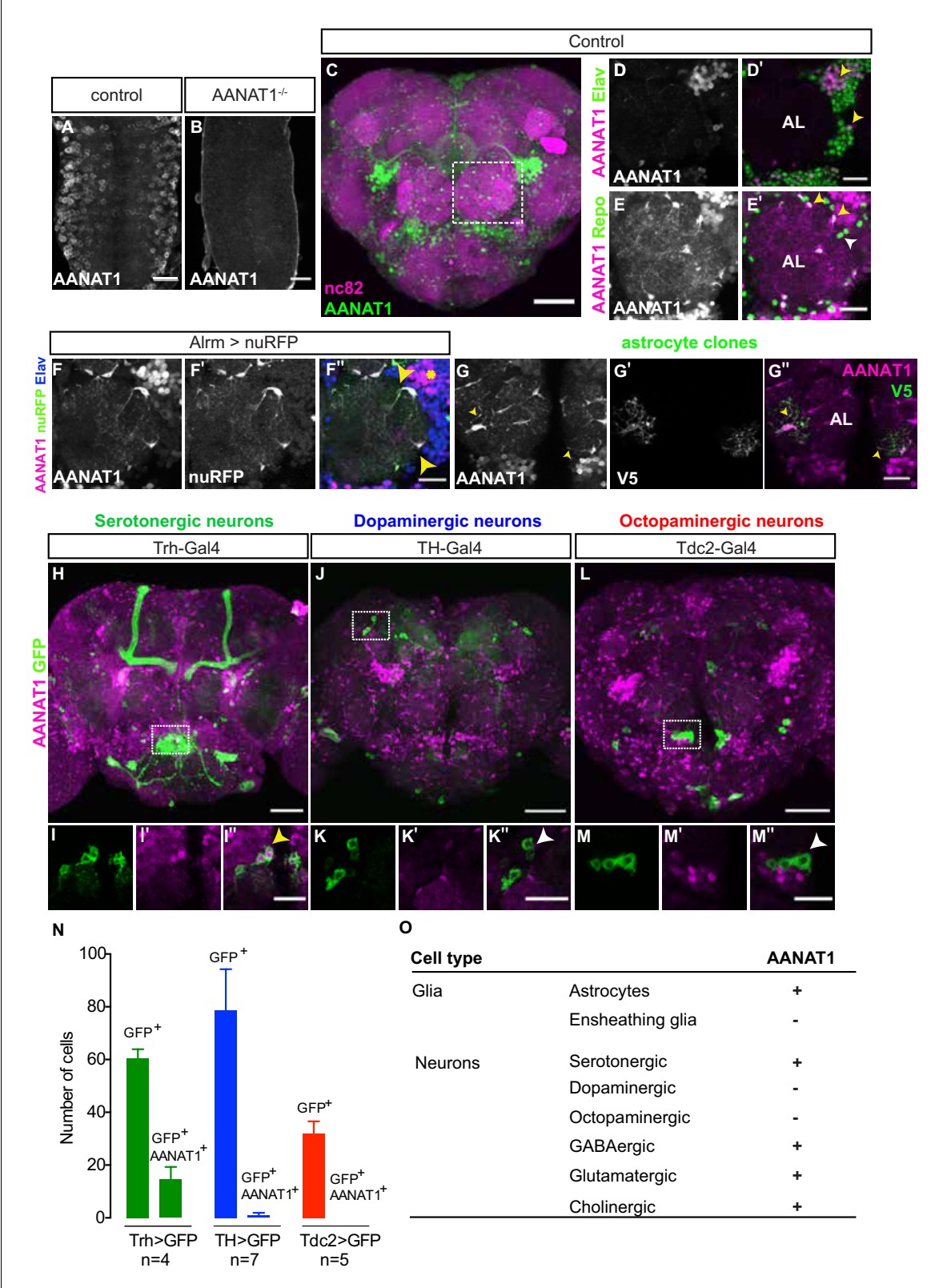

**Figure 1.** AANAT1 expression in the adult *Drosophila* brain. (A–B) AANAT1 IHC in the ventral nerve cords of age-matched embryos (stage 17) of a *w1118* control (A) and an *AANAT1*-null homozygous for Df(BSC)356 (B). (C–M) AANAT1 IHC in the central brain of adults. (C) Z-stack projection showing AANAT1 (green) and neuropil marker nc82 (magenta) in Alrm-Gal4/+ control animals. (D–D') Single optical slice showing AANAT1 (magenta) and the pan-neuronal marker Elav (green). Yellow arrowheads point to neurons co-expressing both (D'). (E–E'). Single optical slice of AANAT1 (magenta) and
*Figure 1 continued on next page*

*Figure 1 continued*

the pan-glial marker Repo (green) in control animals where most glia express AANAT1 (E'; yellow arrowheads), but not all (E'; white arrowheads). AL = antennal lobe. Glia were labeled less intensely for AANAT1 than neurons, and so the adjusted imaging parameters reveal higher background levels in the antennal lobe than seen in D-D'. (**F–F''**) Single slice of AANAT1 (magenta), Elav (blue), and astrocyte marker Alrm-Gal4; UAS-nuRFP (Alrm >nuRFP, green) showing co-expression of AANAT1 and nuRFP in astrocytes (F''; yellow arrowheads) and with Elav (F''; yellow asterisk). (**G–G''**) MCFO-labeled single-cell astrocyte clones (anti-V5, green) co-labeled with AANAT1 (magenta). Yellow arrowheads indicate AANAT1-positive astrocyte cell bodies and cytoplasm. AL = antennal lobe. (**H–M''**) Z-stack projections and single=slice images of AANAT1 (magenta) and GFP (green) IHC in monoaminergic neurons labeled with type-specific Gal4 drivers. Dotted boxes in H, J and L show regions approximating those selected for imaging at higher power in animals of the same genotypes shown in I, K and M, respectively. AANAT1 is expressed in some serotonergic neurons (I''; yellow arrowhead), but not in dopaminergic or octopaminergic neurons (K'', M''; white arrowheads). (**N**) Quantification of the mean number of GFP-positive and GFP/AANAT1 double-positive cells in the central brains of animals where Gal4 is used to express GFP in serotonergic (green), dopaminergic (blue) or octopaminergic (red) neurons. Error bars represent standard deviation. (**O**) Summary of AANAT1 expression in cell types of the adult *Drosophila* central brain. Scale bars in A, B, D-G, I, K, M = 20 µm. Scale bars in C, H, J, L = 50 µm.

The online version of this article includes the following figure supplement(s) for figure 1:

**Figure supplement 1.** AANAT1 expression in the adult *Drosophila* brain.

RNA interference (RNAi)-mediated knockdown of AANAT1 from all neurons using the driver nSyb-Gal4, the AANAT1-positive glial cells in the central brain could be identified more clearly as astrocytes by their ramified morphology, where AANAT1 could be observed in their thin, neuropil-infiltrating processes (*Figure 1—figure supplement 1D,G,H*).Their identity as astrocytes was further confirmed by the morphologies of single cells labeled with the Multi-Color Flp-OUT (MCFO) system (*Figure 1G*). In contrast to astrocytes, AANAT1 expression was absent from ensheathing glia marked by R56F03-Gal4 (*Figure 1—figure supplement 1B*).

Labeling of astrocytes was confirmed to be specific for AANAT1 because it was lost upon knockdown of AANAT1 from astrocytes with Alrm-Gal4 or Repo-Gal4 (*Figure 1—figure supplement 1C, E,I,J*). This also revealed more clearly the several clusters of AANAT1-positive neurons and their axon tracts in the central complex of the brain, which we examined in brain regions associated with sleep regulation (*Figure 1—figure supplement 1C,E*). AANAT1 expression was largely absent from the neuropils of the mushroom body (MB) and fan-shaped body (FSB), though there were scattered AANAT1-positive astrocytes nearby. AANAT1 expression in the neuropil of the ellipsoid body (EB) came almost exclusively from neurons, as revealed by neuron-selective RNAi knockdown (*Figure 1—figure supplement 1K–Q*). Elsewhere, it appeared that astrocytes contributed far more to AANAT1 labeling of brain neuropil regions than did neurons; for example, in the antennal lobe (*Figure 1E,F*) and subesophageal ganglion (*Figure 1—figure supplement 1D*), AANAT1 expression within neuropil regions came primarily from the infiltrative processes of astrocytes.

The monoamines serotonin, dopamine, and octopamine (the insect equivalent of norepinephrine) are known to act in the fly brain to regulate the quantity and timing of sleep (*Nall and Sehgal, 2014*). Pharmacological, genetic, and thermogenetic approaches have converged to demonstrate that serotonin signaling in the fly brain increases sleep, whereas dopamine or octopamine signaling promote waking (*Andretic et al., 2005*; *Artiushin and Sehgal, 2017*; *Kume et al., 2005*; *Nall and Sehgal, 2014*; *Qian et al., 2017*; *Wu et al., 2008*; *Yuan et al., 2005*). Previous studies have suggested AANAT1 to be expressed in dopaminergic neurons (*Ganguly-Fitzgerald et al., 2006*; *Shao et al., 2011*), but this has not been tested directly. With IHC, we examined AANAT1 co-labeling of serotonergic, dopaminergic and octopaminergic neurons using a mCD8-GFP reporter driven by either Trh-Gal4 (*Alekseyenko et al., 2010*), TH-Gal4 (*Friggi-Grelin et al., 2003*), or Tdc2-Gal4 (*Monastirioti et al., 1995*), respectively. AANAT1 was expressed in an average of 14.5 ± 4.8 of 60 ± 3.4 (25%) of serotonergic cells labeled with Trh >mCD8 GFP (*Figure 1H,I–I'',N*), which were found largely in a cluster within the medial subeosophageal ganglion (*Figure 1H*). However, AANAT1 did not co-label cells expressing TH >mCD8 GFP or Tdc2 >mCD8 GFP (*Figure 1J,K–K'', L,M–M'', N*), indicating AANAT1 is not expressed in dopaminergic or octopaminergic neurons. These results are corroborated by single-cell RNA sequencing data showing AANAT1 transcripts in astrocytes and serotonergic neurons (*Croset et al., 2018*).

To identify the other types of neurons expressing AANAT1, we used a mCD8-GFP reporter driven by either MiMIC-vGlut, Gad1-Gal4, or Cha-Gal4 and found AANAT1 in sub-populations of neurons that release glutamate, gamma-aminobutyric acid (GABA), or acetylcholine, respectively (*Figure 1—*

*figure supplement 1R-W''*). Monoamines are mainly synthesized in the neurons that release them, and it is generally understood that their re-uptake into these same neurons occurs via specific transport proteins to prevent their accumulation at synapses (*Martin and Krantz, 2014*). Absence of AANAT1 from dopaminergic or octopaminergic neurons showed that cells that produce and release monoamines do not necessarily contribute to their catabolism via AANAT1. However, the presence of AANAT1 in subsets of glutamatergic, GABAergic and cholinergic neurons suggests that, along with astrocytes, these non-monoaminergic neurons could contribute to regulation of monoamine bioavailability in the brain.

The *AANAT1* gene produces two protein isoforms, the shorter of which (FlyBase AANAT1-PA, 240aa in length), previously known as aaNAT1b, is more predominant (*Brodbeck et al., 1998*). This shorter isoform was observed to be lost in *AANAT1^{lo}* mutants (*Hintermann et al., 1996*). *AANAT1^{lo}* is a spontaneous mutation that arose from insertion of a transposable element into the *AANAT1* gene, and tissue extracts from these flies have reduced AANAT1 activity (*Hintermann et al., 1996*; *Maranda and Hodgetts, 1977*). Using our new AANAT1 antiserum to perform western blotting of brain extracts, we observed only the shorter isoform in controls (*Figure 2A*). In *AANAT1^{lo}* homozygotes and hemizygotes (*AANAT1^{lo}/In(2LR)Px[4]*), AANAT1 protein levels were reduced to 13 and 8% of iso31 controls, respectively (*Figure 2A,B*). This was confirmed with IHC in the brains of *AANAT1^{lo}* flies (*Figure 2C–E*), where we noted residual AANAT1 expression in some Elav[+] neurons, but complete loss of AANAT1 from astrocytes (*Figure 2F–F', G–G'*).

In vitro studies have shown that serotonin and dopamine are substrates for AANAT1 with similar affinities (*Hintermann et al., 1995*). Whether the levels of these and/or other monoamines are regulated by AANAT1 in vivo remains to be determined. We used High Performance Liquid Chromatography - Mass Spectrometry (HPLC-MS) to measure levels of serotonin, dopamine, and octopamine in the brains of *AANAT1^{lo}* flies and controls (iso31) (*Figure 2H*). Under baseline sleep-cycle conditions, where brain tissues were collected in a 3 hr window after lights-ON (ZT0), serotonin and dopamine levels in *AANAT1^{lo}* flies were similar to controls (*Figure 2I*). Octopamine was undetectable in controls and was found at low levels in brains of *AANAT1^{lo}* flies (*Figure 2—figure supplement 1B*). However, if this window was preceded by 12 hr (ZT12-ZT24) of SD overnight, *AANAT1^{lo}* brains had a robust increase in the levels of serotonin and dopamine compared to controls (*Figure 2I*), but this had no effect on octopamine levels (*Figure 2—figure supplement 1B*). Importantly, in control animals the SD protocol itself did not appear to affect the levels of measured monoamines, or of AANAT1 itself (*Figure 2J,K*). Further, we did not observe changes in the levels of another monoamine catabolic enzyme known to be expressed in astrocytes (Ebony) in either *AANAT1^{lo}* flies, or in flies subjected to SD (*Figure 2—figure supplement 1A,C*). We conclude that catabolism of serotonin and dopamine in the brains of flies lacking AANAT1 is severely compromised upon SD, leading to inappropriate accumulation of these monoamines.

The *AANAT1^{lo}* mutation increases homeostatic sleep following deprivation (*Shaw et al., 2000*), suggesting AANAT1 could be key to how the brain limits the homeostatic response to sleep need. *AANAT1^{lo}* is also interesting because these flies were reported to have normal motor activity, and intact daily patterns of sleep (*Shaw et al., 2000*), allowing genetic dissection of homeostatic sleep-control independent of the regulation of baseline sleep. We wondered whether the increased recovery sleep seen in *AANAT1^{lo}* animals could be explained by loss of AANAT1 function from neurons or astrocytes. To test this, we selectively knocked down AANAT1 in distinct cell types with RNAi and measured both baseline and homeostatic sleep with the *Drosophila* Activity Monitoring System (DAMS). To evaluate the contribution of neuronal AANAT1 to sleep, we tested nSyb-Gal4 >UAS-AANAT1-RNAi flies, using two independent RNAi lines that target the AANAT1 transcript at distinct sites. Knockdown of AANAT1 in neurons with either line resulted in normal patterns of baseline sleep (*Figure 3A–H*), as reported for the *AANAT1^{lo}* allele (*Shaw et al., 2000*). While awake during daytime, these animals had lower levels of activity than controls carrying either the GAL4 or UAS transgene alone (*Figure 3—figure supplement 1A,B*), but their total daytime sleep amount was similar to at least one of the controls (*Figure 3B,F*), as were the length of daytime sleep bouts (*Figure 3C,G*) and their number (*Figure 3D,H*). When we examined the amount of nighttime sleep compared to controls, we found that one RNAi line (AANAT-RNAi 2, JF02142), but not the other (AANAT-RNAi 1, HMS01617), led to increased amount of nighttime sleep (*Figure 3B,F*). For knockdown with this RNAi line only, sleep bouts during the night were increased in duration and decreased in number, suggesting improved sleep consolidation at night (*Figure 3C–D;G-H*). We

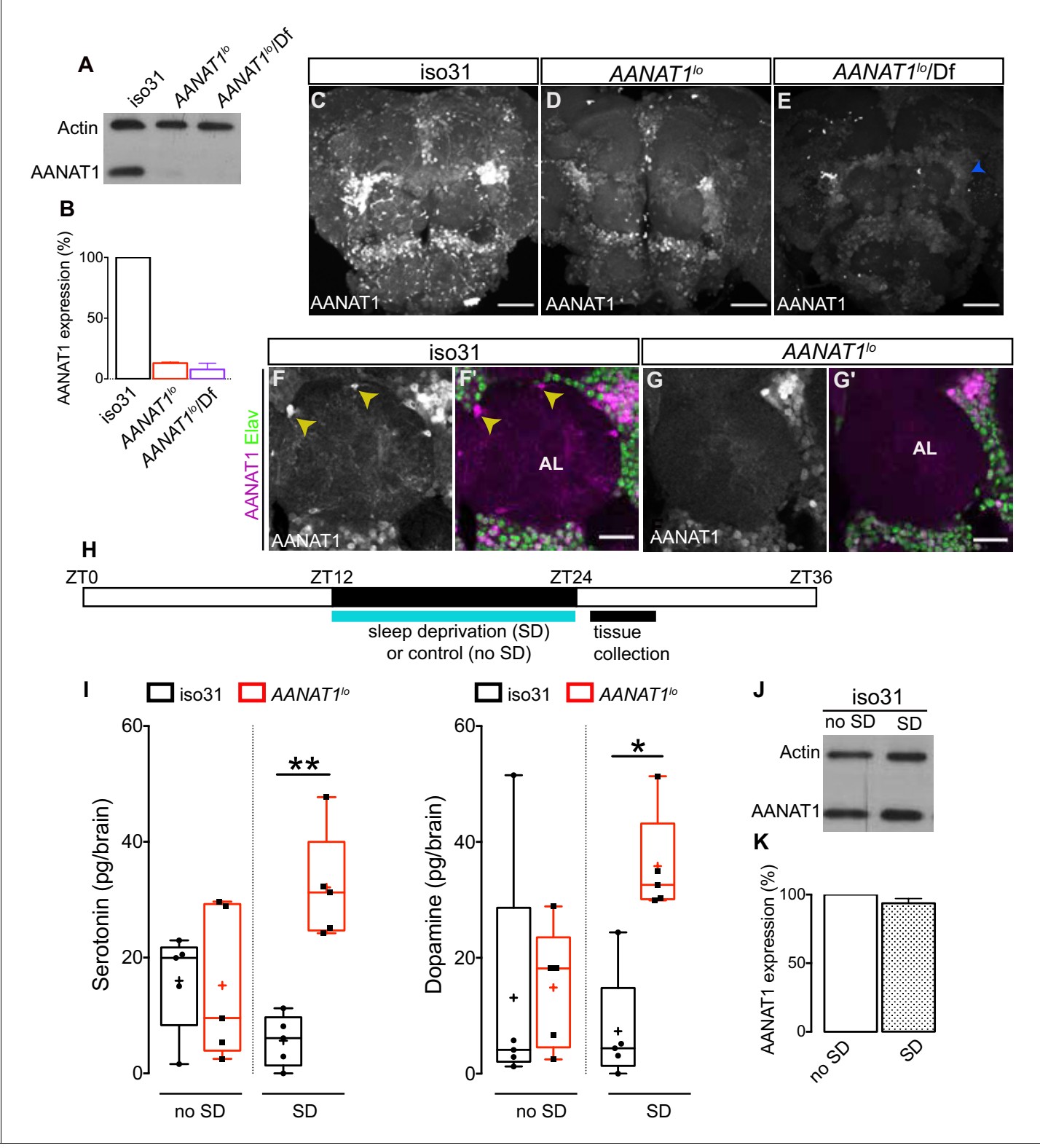

**Figure 2.** Characterization of *AANAT1^lo^*. (**A**) Western blot of lysates prepared from dissected brains (ZT9-10) of iso31, *AANAT1^lo^* and *AANAT1^lo^*/Df(In (2LR)Px[4]) adult males. (**B**) Quantification of AANAT1 expression normalized to that of actin (mean + standard deviation, n = 3 biological replicates). (**C–E**) Z-stack images showing AANAT1 (gray) in iso31 (**C**), *AANAT1^lo^* (**D**) and *AANAT1^lo^*/Df(In(2LR)Px[4]) (**E**) animals. Blue arrowhead in E represents background signal. Scale bars = 50 μm. (**F, G**) Single optical slices showing AANAT1 (gray or magenta) and Elav (green) in iso31 (**F, F'**) and *AANAT1^lo^* (**G, G'**). Yellow arrowhead shows AANAT1^+^ astrocytes. Scale bars = 20 μm. (**H**) Schematic of experiment for HPLC-MS analysis. (**I**) HPLC-MS

*Figure 2 continued on next page*

Figure 2 continued

measurement of serotonin (one-way ANOVA with Tukey's post-hoc test, *p<0.05, **p<0.01,) and dopamine (Kruskal–Wallis test, Dunn's multiple comparisons, *p<0.05,) in iso31 (black) and *AANAT1^{lo}* (red) fly brains under control and sleep deprivation (SD) conditions. Box and whisker plots show 25–75% interquartile range (box), minimum and maximum (whiskers), median (horizontal line in box), and mean (+). n = 5 per genotype. (J) Western blot of lysates prepared from dissected brains (ZT24-25) of iso31 females in control (no SD) and SD conditions. (K) Quantification of AANAT1 (paired t-test, p=0.0831, n = 3) expression, normalized to actin levels in iso31 animals under control (no SD) and SD conditions.
The online version of this article includes the following figure supplement(s) for figure 2:

**Figure supplement 1.** Characterization of *AANAT1^{lo}*.

then assessed whether AANAT1 knockdown in neurons (nSyb >AANAT1 RNAi) would impact sleep recovery after SD, as was observed in *AANAT1^{lo}* flies. For this, flies were subjected to overnight mechanical SD and we found, somewhat surprisingly, that these flies did not display enhanced recovery sleep the next day (*Figure 3I–L*).

Next, we used Alrm-Gal4 to selectively deplete AANAT1 expression from astrocytes with RNAi (Alrm >AANAT1 RNAi). This had no effect on the numbers of astrocytes present in the brain (*Figure 4A*). These flies showed normal baseline patterns and amounts of daytime and nighttime sleep compared to controls (*Figure 4B–E*), but while awake they were less active than controls (*Figure 4—figure supplement 1A,B*). However, upon overnight mechanical SD, these flies had increased recovery sleep the next day (*Figure 4F–I*), mimicking *AANAT1^{lo}* flies (*Figure 4—figure supplement 1C–E*). Like AANAT1 loss-of-function, AANAT1 overexpression in astrocytes also increased recovery sleep following deprivation (*Figure 4—figure supplement 1F*), underscoring the importance of regulated astrocytic AANAT1 levels in sleep homeostasis.

We characterized AANAT1 expression in astrocytes during pupal development with immunochemistry, and found AANAT1 to be expressed weakly in only a few astrocytes at 48 hr after puparium formation (APF), then gradually more strongly in most but not all astrocytes at 72 hr and 96 hr APF (*Figure 4—figure supplement 1G-I''*). To investigate when AANAT1 functions in astrocytes for sleep recovery, we used the Temporal And Regional Gene Targeting (TARGET) system (*McGuire et al., 2004*) to knock down AANAT1 in adult astrocytes with Eaat1-Gal4. In the brain, Eaat1-Gal4 is a driver line for astrocytes (which express AANAT1) and cortex glia (which do not). When adult flies were raised at 32°C to deplete AANAT1 from glia using RNAi, these animals showed increased recovery sleep compared to the UAS control but not the Gal4 control (*Figure 4—figure supplement 1J-K*).

Our results demonstrate that AANAT1 acts in astrocytes, but not in neurons, to restrict daytime recovery sleep in response to overnight SD. With HPLC-MS, we noted that astrocyte-selective AANAT1 knockdown led to increased levels of brain serotonin and dopamine after SD in most samples compared to controls (*Figure 4J*). This did not reach statistical significance however, perhaps because of an outlier in each of the control groups, or perhaps because AANAT1 knockdown in astrocytes affects the levels of only a portion of serotonin and dopamine in the brain, albeit an important portion with respect to sleep homeostasis. Together with the clear increases seen in *AANAT1^{lo}* flies (*Figure 2I*), these data suggest that *Drosophila* astrocytes employ AANAT1 to limit accumulation of serotonin and dopamine upon SD.

Since loss of AANAT1 does not affect astrocyte numbers or their terminal differentiation, we think it unlikely to play a developmental role and favor the idea that AANAT1 functions in mature astrocytes to limit recovery sleep. To shed light on when AANAT1 might be active with respect to sleep homeostasis, with IHC we examined AANAT1 under baseline sleep-cycle conditions at 3 hr intervals during the light and dark period. In the dark period ZT12-ZT24, the patterns of AANAT1 expression in sleep regulatory regions the MB, FSB and EB were unchanged compared to the light period patterns described above (data not shown). In addition, we never observed obvious changes of AANAT1 levels in astrocytes over the course of the light period ZT0-ZT12. Interestingly, during the dark phase most astrocytes throughout the brain showed obvious changes of AANAT1 levels. Dark period AANAT1 expression in astrocyte cell bodies peaked at ZT15, declined markedly to undetectable levels by ZT21 (*Figure 4K*), and was re-established at lights-ON (ZT24). From this we speculate that the loss of AANAT1 might have profound influence on sleep homeostasis near ZT15, when

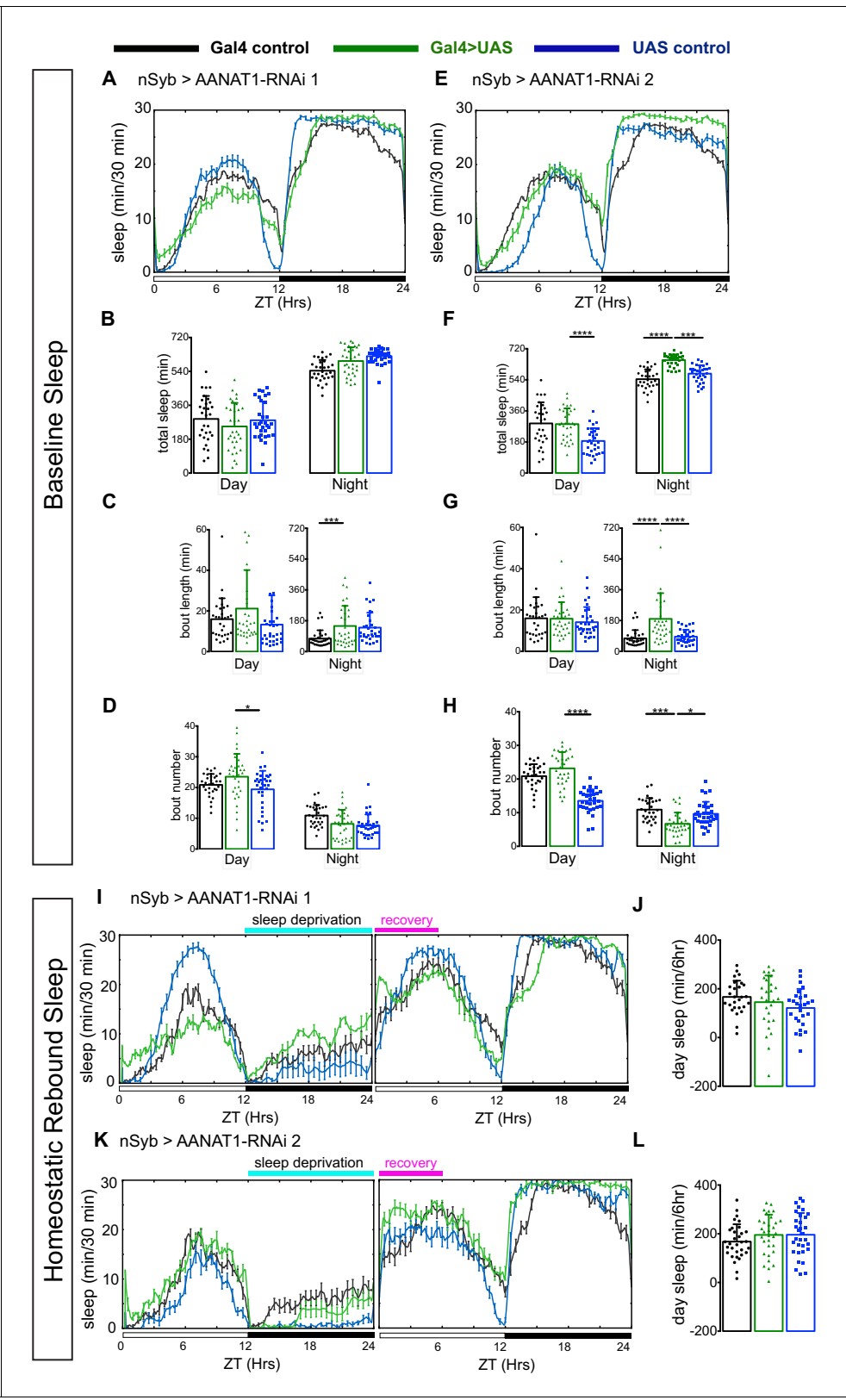

**Figure 3.** AANAT1 knockdown in neurons. (A–D) Baseline sleep upon knockdown with UAS-HMS01617 (RNAi 1). 24 hr sleep profile showing light/dark conditions on X-axis (A), and quantification during day (ZT 0–12) versus night (ZT 12–24) of total sleep duration (B), sleep bout length (C) and bout number (D) for the nSyb-Gal4 control (black, n = 30), the UAS-HMS01617 control (blue, n = 32), and knockdown animals (nSyb >HMS01617, green, n = 32). (bar graphs show mean + standard deviation, one-way ANOVA with Tukey's post-hoc test, *p<0.05, ***p<0.001, ****p<0.0001). (E–H) Baseline

*Figure 3 continued on next page*

*Figure 3 continued*

sleep upon knockdown with UAS-JF02142 (RNAi 2). (**E**) 24 hr sleep profile showing light/dark conditions on X-axis. Quantification of total sleep duration (**F**), sleep bout length (**G**) and bout number (**H**) for the nSyb-Gal4 control (black, n = 30), the UAS- JF02142 control (blue, n = 32), and knockdown animals (nSyb >JF02142, green, n = 32). The plotted nSyb-Gal4 control data is the same as in A-D, as the experiments were done simultaneously. (one-way ANOVA with Tukey's post-hoc test, \*\*\*p<0.001, \*\*\*\*p<0.0001). (**I, J**) Recovery sleep upon knockdown with UAS-HMS01617 (RNAi 1). (**I**) 24 hr sleep profile of baseline and recovery days, and (**J**) the duration of sleep during ZT0-6 recovery period. nSyb-Gal4 control (black), the UAS-HMS01617 control (blue), and knockdown animals (nSyb >HMS01617, green). (n = 27 per genotype, one-way ANOVA with Tukey's post-hoc test). (**K, L**) Recovery sleep upon knockdown with UAS-JF02142 (RNAi 2). (**K**) 24 hr sleep profile of baseline and recovery day, (**L**) duration of sleep during ZT0-6 recovery period (**L**). nSyb-Gal4 control (black), the UAS- JF02142 control (blue), and knockdown animals (nSyb >HMS01617, green). (n = 32 per genotype, one-way ANOVA with Tukey's post-hoc test).

The online version of this article includes the following figure supplement(s) for figure 3:

**Figure supplement 1.** AANAT1 knockdown in neurons.

AANAT1 levels in astrocytes are usually highest, or during daytime when the increased recovery sleep occurs.

AANAT1 is expressed in astrocytes that reside throughout the brain, and so it remains unclear whether it modulates sleep homeostasis by acting within a particular region of the brain, or more broadly. Interestingly, the neuropils of key sleep centers (MB, EB and FSB; *Figure 1—figure supplement 1K–Q*) had no AANAT1 staining from infiltrative astrocytes, raising the likelihood it acts elsewhere. Finally, it remains to be established whether the effect of AANAT1 on sleep homeostasis is due to serotonin, dopamine, or both. In *Drosophila*, serotonergic signaling in the brain promotes sleep, while dopaminergic signaling promotes waking. Levels of both serotonin and dopamine are upregulated in *AANAT1^{lo}* mutants upon SD, where increased sleep prevails. It stands to reason that AANAT1 could act in astrocytes to limit the deprivation-dependent accumulation of sleep-promoting serotonin. It is also possible that dopamine accumulation plays a role, since thermogenetic activation of wake-promoting dopaminergic neurons at night promotes compensatory sleep the next day. This suggests these particular neurons are upstream of circuits that produce homeostatic responses to extended wakefulness (*Dubowy and Sehgal, 2017*; *Seidner et al., 2015*), and astrocytic AANAT1 could somehow restrict dopaminergic signaling from these neurons overnight.

Our findings illustrate a newly discovered role for astrocytes in the control of monoamine bioavailability and homeostatic sleep drive, where they are specifically engaged to catabolize monoamines whose levels are elevated by overnight SD. *Drosophila* astrocytes also express the enzyme Ebony, which couples dopamine to N-β-alanine (*Suh and Jackson, 2007*), and a receptor for octopamine and tyramine (*Ma et al., 2016*), reinforcing how they are well-equipped to metabolize monoamines, and to monitor and respond to monoaminergic neuronal activity. Neither gene expression studies nor RNA sequencing databases provide evidence for monoamine-synthesizing enzymes in *Drosophila* astrocytes, so it appears likely that monoamines inactivated by AANAT1 in astrocytes are brought into these cells by an unidentified transporter. Astrocytes are particularly well-suited for regulating sleep in this way because they have ramified processes that infiltrate neuropil regions to lie in close proximity to synapses. SD can seemingly reduce the degree of contact between astrocytes and neurons in the fly brain (*Vanderheyden et al., 2019*), and so it is possible that these structural changes could influence monoamine uptake and inactivation by astrocytes.

In neurons, AANAT1 may function to limit sleep consolidation at night, but evidence for this came from only one of the two RNAi lines used in this study and was not observed in *AANAT1^{lo}* mutants (*Figure 4—figure supplement 1A*). Further studies are needed to characterize sleep-control functions of AANAT1 in neurons, if any, to understand better how cellular context can impact AANAT1 function in sleep regulation. In light of this, we note that loss of the related enzyme *aanat2* in zebrafish larvae decreases baseline sleep (*Gandhi et al., 2015*), which could be attributed to a loss of melatonin since the AANAT1 product N-acetylserotonin is an intermediate in the synthesis of melatonin in vertebrates (*Hintermann et al., 1996*). Clearly, the appropriate balance and cellular context of AANAT activity is critical for the regulation of sleep, and we show here in *Drosophila* that astrocytes are an important contributor to this balance. Interestingly, astrocytes in rodents express the monoamine transporters and receptors for dopamine and serotonin (*Bacq et al., 2012*; *Baganz et al., 2008*; *Huang et al., 2012*; *Petrelli et al., 2020*; *Sandén et al., 2000*;

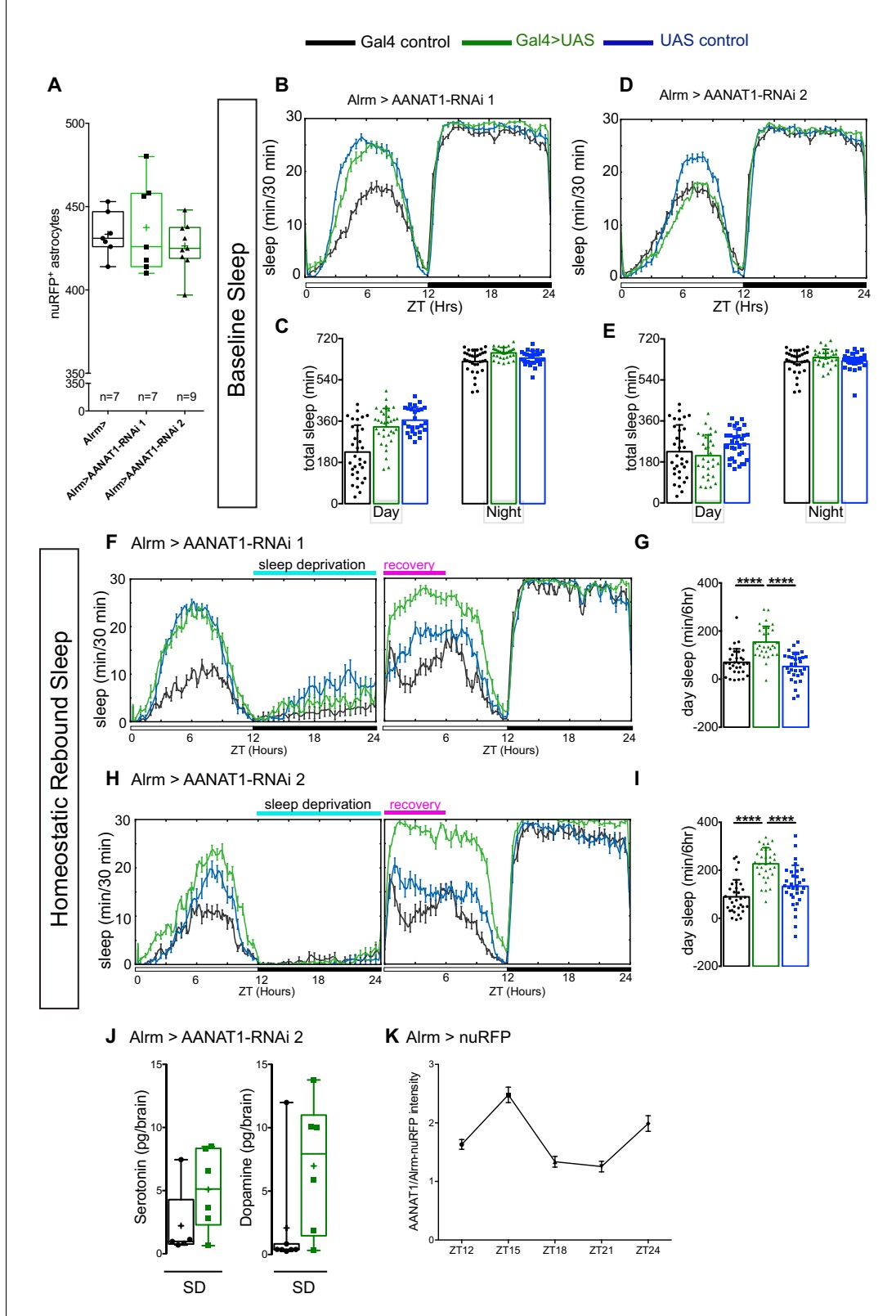

**Figure 4.** AANAT1 knockdown in astrocytes. (**A**) Compared with Alrm-Gal4 controls (Alrm>), the number of nuRFP labeled astrocytes in the central brain is unaffected upon RNAi knockdown of AANAT1 with HMS01617 (AANAT1-RNAi 1) or JF02142 (AANAT1-RNAi 2). Box and whisker plot as in *Figure 2F*. One-way ANOVA with Tukey's post-hoc test, n = 7–9 per genotype. (**B, C**) Baseline sleep upon knockdown with UAS-HMS01617 (RNAi 1). 24 hr sleep profile (**B**), and total sleep duration (**C**) for the Alrm-Gal4 control (black; n = 32), the UAS-HMS01617 control (blue; n = 26), and knockdown

*Figure 4 continued*

animals (Alrm >HMS01617, green; n = 32), (one-way ANOVA with Tukey's post-hoc test). (**D, E**) Baseline sleep upon knockdown with UAS-JF02142 (RNAi 2). 24 hr sleep profile (**D**), and total sleep duration (**E**) for the Alrm-Gal4 control (black; n = 32), the UAS-JF02142 control (blue; n = 32), and knockdown animals (Alrm >JF02142, green; n = 30). The plotted Alrm-Gal4 control data is the same as in B and C, as the experiments were done simultaneously, (one-way ANOVA with Tukey's post-hoc test). (**F, G**) Recovery sleep upon knockdown with UAS-HMS01617 (RNAi 1). 24 hr sleep profile of baseline day and recovery day (**F**), and the duration of sleep during ZT0-6 recovery period (**G**) for the Alrm-Gal4 control (black), the UAS-HMS01617 control (blue), and knockdown animals (Alrm >HMS01617, green). (n = 31 per genotype, Kruskal–Wallis one-way ANOVA with Dunn's post-hoc test, ****p<0.0001). (**H, I**) Recovery sleep upon knockdown with UAS-JF02142 (RNAi 2). 24 hr sleep profile of baseline day and recovery day (**H**), and the duration of sleep during ZT0-6 recovery period (**I**) for the Alrm-Gal4 control (black), the UAS-JF02142 control (blue), and knockdown animals (Alrm >HMS01617, green). (n = 32 per genotype, error bars are mean + standard deviation, one-way ANOVA with Tukey's post-hoc test, ****p<0.0001). (**J**) HPLC-MS measurement of serotonin (Mann-Whitney t-test) and dopamine (Mann-Whitney t-test) in Alrm-Gal4 control (black; n = 7 for dopamine; n = 5 for serotonin) and Alrm >JF02142 (green; n = 6) fly brains under sleep deprivation (SD) conditions. For UAS controls (not shown), some samples fell below the limit of detection, leaving too few data points for robust statistical analysis. Box and whisker plots show 25–75% interquartile range (box), minimum and maximum (whiskers), median (horizontal line in box), and mean (+). (**K**) AANAT1 levels in astrocyte cell bodies normalized to nuRFP at ZT12,15,18,21 and 24 time-points. (n = 3 per time-point, 10 cells per sample, mean+ SEM).

The online version of this article includes the following figure supplement(s) for figure 4:

**Figure supplement 1.** AANAT1 knockdown in astrocytes.

*Vaarmann et al., 2010*), raising the possibility that astrocytes in mammals might also participate in mechanisms of sleep regulation involving monoaminergic neural signaling.

# Materials and methods

**Key resources table**

| Reagent type (species) or resource | Designation | Source or reference | Identifiers | Additional information |
|---|---|---|---|---|
| Gene (*Drosophila melanogaster*) | AANAT1 | Flybase | FBgn0019643 | Previously known as Dat. Also, speck (sp) |
| Genetic reagent (*D. melanogaster*) | Trh-Gal4 | Bloomington *Drosophila* Stock Center | RRID:BDSC_52249 | FlyBase genotype: y$^1$ w*; wgSp-1/CyO, P{Dfd-EYFP}2; P{Trh-GAL4.S}attP2 |
| Genetic reagent (*D. melanogaster*) | TH-Gal4 | Bloomington *Drosophila* Stock Center | RRID:BDSC_8848 | FlyBase genotype: w*; P{ple-GAL4.F}3 |
| Genetic reagent (*D. melanogaster*) | Tdc2-Gal4 | Bloomington *Drosophila* Stock Center | RRID:BDSC_52243 | FlyBase genotype: y$^1$ w*; wg$^{Sp-1}$/CyO, P{Dfd-EYFP}2; P{Tdc2-GAL4.S}attP2 |
| Genetic reagent (*D. melanogaster*) | 10X-UAS-mCD8-GFP | Bloomington *Drosophila* Stock Center | RRID:BDSC_32186 | FlyBase genotype: w*; P{10XUAS-IVS-mCD8::GFP}attP40 |
| Genetic reagent (*D. melanogaster*) | UAS-RFP.nls | Bloomington *Drosophila* Stock Center | RRID:BDSC_30558 | FlyBase genotype: w[1118]; P{w[+mC]=GAL4-Act5C(FRT.CD2).P}S, P{w[+mC]=UAS RFP.W}3/TM3, Sb[1] |
| Genetic reagent (*D. melanogaster*) | Mi{MIC} VGlut$^{MI04979}$ | Bloomington *Drosophila* Stock Center | RRID:BDSC_38078 | FlyBase genotype: y$^1$ w*; Mi{MIC}VGlut$^{MI04979}$ |
| Genetic reagent (*D. melanogaster*) | Gad1-Gal4 | Bloomington *Drosophila* Stock Center | RRID:BDSC_51630 | FlyBase genotype: P{Gad1-GAL4.3.098}2/CyO |
| Genetic reagent (*D. melanogaster*) | Cha-Gal4 | Bloomington *Drosophila* Stock Center | RRID:BDSC_6793 | FlyBase genotype: w*; P{ChAT-GAL4.7.4}19B P{UAS-GFP.S65T}Myo31DF$^{T2}$ |
| Genetic reagent (*D. melanogaster*) | R56F03-Gal4 | Bloomington *Drosophila* Stock Center | RRID:BDSC_39157 | FlyBase genotype: w[1118]; P{y[+t7.7] w[+mC]=GMR56 F03-GAL4}attP2 |

*Continued on next page*

*Continued*

| Reagent type (species) or resource | Designation | Source or reference | Identifiers | Additional information |
|---|---|---|---|---|
| Genetic reagent (*D. melanogaster*) | *AANAT1^lo* | Bloomington *Drosophila* Stock Center | RRID:BDSC_3193 | FlyBase genotype: bw[1] AANAT1[lo] |
| Genetic reagent (*D. melanogaster*) | Df(2R)BSC356 | Bloomington *Drosophila* Stock Center | RRID:BDSC_24380 | FlyBase genotype: w[1118]; Df(2R)BSC356/SM6a |
| Genetic reagent (*D. melanogaster*) | In(2LR)Px4 | Bloomington *Drosophila* Stock Center | RRID:BDSC_1473 | FlyBase genotype: In(2LR)Px[4], dpy[ov1] b[1]/CyO |
| Genetic reagent (*D. melanogaster*) | tubGal80^ts | Bloomington *Drosophila* Stock Center | RRID:BDSC_7018 | FlyBase genotype: w[*]; sna[Sco]/CyO; P{w[+mC]=tubP-GAL80[ts]} ncd[GAL80ts-7] |
| Genetic reagent (*D. melanogaster*) | UAS-HMS01617 | Bloomington *Drosophila* Stock Center | RRID:BDSC_36726 | FlyBase genotype: y[1] sc[*] v[1] sev[21]; P{y[+t7.7] v[+t1.8]=TRiP.HMS01617}attP40/CyO |
| Genetic reagent (*D. melanogaster*) | UAS-JF02142 | Bloomington *Drosophila* Stock Center | RRID:BDSC_26243 | FlyBase genotype: y[1] v[1]; P{y[+t7.7] v[+t1.8]=TRiP. JF02142}attP2 |
| Genetic reagent (*D. melanogaster*) | *MCFO* | Bloomington *Drosophila* Stock Center | RRID:BDSC_64085 | FlyBase genotype: FlpG5.Pest; 10xUAS(FRT-stop)myr::smGdP-HA, 10xUAS(FRT-stop)myr::smGdP-V5-THS-10xUAS(FRT-stop)myr::smGdP-FLAG |
| Genetic reagent (*D. melanogaster*) | Alrm-Gal4 | Marc Freeman | | |
| Genetic reagent (*D. melanogaster*) | Eaat1-Gal4 | Marc Freeman | | |
| Genetic reagent (*D. melanogaster*) | nSyb-Gal4 | Julie Simpson, Stefan Thor | | |
| Genetic reagent (*D. melanogaster*) | UAS-AANAT1 | This paper | | See Materials and methods: Fly stocks |
| Cell lines (*Escherichia coli*) | GH12636 | *Drosophila* Genomic Research Centre | FBcl0129063 | |
| Cell lines (*Escherichia coli*) | pJFRC-MUH | *Addgene* | RRID:Addgene_26213 | |
| Antibody | AANAT1 (Rabbit polyclonal) | MEDIMABS, Montreal | This paper | IHC(1:2000), WB (1:2500). See Materials and methods: Generation of AANAT1 antibody |
| Antibody | Elav (Rat monoclonal) | Developmental Studies Hybridoma Bank | Elav-7E8A10 | IHC(1:100) |
| Antibody | Repo (Mouse monoclonal) | Developmental Studies Hybridoma Bank | 8D12 | IHC(1:50) |
| Antibody | Brp (Mouse monoclonal) | Developmental Studies Hybridoma Bank | nc82 | IHC(1:50) |
| Antibody | GFP (Mouse monoclonal) | Clontech | #632381 | IHC(1:200) |
| Antibody | Ebony (Rabbit polyclonal) | Sean Carroll | | WB (1:3000) |
| Antibody | Actin (Mouse monoclonal) | Sigma | #A4700 | WB (1:3000) |
| Commercial assay or kit | HyGLO Chemiluminescent HRP Antibody Detection Reagent | Denville Scientific | | |
| Chemical compound, drug | Formic acid | Fisher scientific | | |

*Continued on next page*

*Continued*

| Reagent type (species) or resource | Designation | Source or reference | Identifiers | Additional information |
|---|---|---|---|---|
| Software, algorithm | Graphpad Prism 6 | | RRID:SCR_002798 | |
| Software, algorithm | Fiji | | RRID:SCR_002285 | |

## Fly stocks

*Drosophila melanogaster* stocks were obtained from the Bloomington *Drosophila* Stock Center (BSC): Trh-Gal4 (BSC-52249), TH-Gal4 (BSC-8848), Tdc2-Gal4 (BSC-9313), Ddc1-Gal4 (BSC-7010), UAS-mCD8-GFP (BSC-32186), UAS-RFP.nls (BSC-30558), Mi{MIC} VGlut$^{MI04979}$ (BSC-38078), Gad1-Gal4 (BSC-51630), Cha-Gal4 (BSC-6793), R56F03-Gal4 (BSC-39157), *AANAT1$^{lo}$* (BSC-3193), Df(2R) BSC356 (BSC-24380), deficiency In(2LR)Px4 (BSC-1473), tubGal80$^{ts}$ (BSC-7018), AANAT1 RNAi lines UAS-HMS01617 (BSC-36726), UAS-JF02142 (BSC-26243) and MCFO stock hs-FlpG5.Pest; 10xUAS (FRT-stop)myr::smGdP-HA, 10xUAS(FRT-stop)myr::smGdP-V5-THS-10xUAS(FRT-stop)myr::smGdP-FLAG (BSC-64085). Alrm-Gal4 and Eaat1-Gal4 was provided by Dr. Marc Freeman, and nSyb-Gal4 by Dr. Stefan Thor.

For RNAi-mediated knockdown of gene expression, control animals carried only a Gal4 driver, while experimental groups also carried a single copy of the transgene to elicit RNAi. The chromosome carrying Alrm-Gal4 also bore a transgene encoding the nuclear reporter UAS-nuRFP. To mitigate the effects of genetic background for sleep experiments, control Gal4 and UAS flies were crossed to the iso31 stock.

In using the TARGET system, we combined GAL80$^{ts}$, a temperature-sensitive inhibitor of GAL4, with EAAT1-GAL4 to selectively knock down AANAT1 (UAS-AANAT1-RNAi 1 (HMS01617)) during adulthood. Animals were raised at the permissive temperature (18°C) to repress Gal4, then 4-day-old adult animals were shifted to 32°C for another 5 days to induce RNAi for AANAT1, before exposing them to SD experiments and sleep monitoring as outlined below.

The morphology of single astrocytes was determined by the MCFO technique (*Nern et al., 2015*), where three differently tagged reporters under UAS control (HA, FLAG and V5) were silenced by FRT-flanked transcriptional terminators. Heat shock-induced FLPase expression removed terminators randomly in individual cells, driven by astrocyte-specific Alrm-GAL4. This created a mosaic of astrocytes of distinct colors. For this experiment, 3–5 days old flies raised at 18°C were heat-shocked at 37°C for 5–8 min and dissected 2–3 days later.

To create UAS-AANAT1, the AANAT1 coding sequence from the cDNA clone GH12636 (*Drosophila* Genomic Research Centre) was PCR-amplified and cloned in-frame into a modified pJFRC-MUH vector (*Pfeiffer et al., 2010*). Transgenic flies with site-specific insertions at *VK0005* site on chromosome three were generated using standard microinjection (BestGene, Inc).

## Generation of AANAT1 antibody

A KLH-coupled peptide RRPSPDDVPEKAADSC (amino acids (aa) 94–109 of isoform AANAT1-PA (FlyBase), or 129–144 of isoform AANAT1-PB) was synthesized and injected into rabbits according to guidelines of the Canadian Council for Animal Care (MEDIMABS, Montreal, QC).

## Immunohistochemistry and imaging

Adult fly brains were dissected between ZT3-9 (unless specified otherwise) in cold phosphate-buffered saline (pH 7.4) and fixed in 4% paraformaldehyde for 30 min (min). After three washes of 15 min each with PBS containing 0.3% Triton-X-100 (PBTx-0.3%), the tissues were blocked in 5% normal goat serum (Jackson Laboratories) in PBTx-0.5% for 45 min. Tissues were incubated in primary antibodies: rabbit anti-AANAT1 (1:2000; this study), rat anti-Elav (1:100; Developmental Studies Hybridoma Bank (DSHB), mouse anti-Repo (1:50; DSHB), mouse anti-nc82 (1:50; DSHB), mouse anti-GFP (1:200; Clontech #632381) overnight at 4°C. After three washes (15 min each, PBTx-0.3%), tissues were incubated with secondary antibodies overnight at 4°C: goat anti-mouse (Rhodamine Red-X, Jackson ImmunoResearch #115-295-146), goat anti-rabbit (Alexa Fluor 488, Thermo Fisher Scientific, #A11008), goat anti-mouse (Alexa Fluor 488, Thermo Fisher Scientific), goat anti-rat (Alexa Fluor

568, Thermo Fisher Scientific, #A11077), goat anti-rabbit (Alexa Fluor 647, Thermo Fisher Scientific, #A21245). Tissues were again washed (3 × 15 min, PBTx-0.3%), followed by a final wash in PBS. Tissues were mounted in SlowFade Diamond Antifade Mountant (Thermo Fisher Scientific, #S36964). Fluorescence images were acquired with an Olympus BX-63 Fluoview FV1000 confocal laser-scanning microscope and processed using Fiji.

For MCFO labeling, brains were dissected in ice-cold PBS, fixed with 4% paraformaldehyde/PBS for 1 hr at room temperature followed by three successive washes in 0.5% PBTx for 20 min each. Simultaneous incubation (48 hr at 4°C) with rat anti-FLAG (1:100; Novus Biologicals NBP1-06712,A-4) and rabbit anti-AANAT1 was followed by another 48 hr at 4°C with goat anti-rabbit (1:1000; Alexa Fluor 488, Thermo Fisher Scientific, #A11008), goat anti-rat (1:1000; Alexa Fluor 568, #A11077) and V5-tag:AlexaFluor-647 (1:200; Bio-Rad MCA1360A647).

To quantify cells immuno-labeled for GFP and AANAT1, cells were manually counted from image stacks of the central brain near the antennal lobe and central complex regions (excluding optic lobes). We chose cell bodies in this dorsal - anterior region because it routinely showed excellent immunochemical signal and good cellular resolution.

## Western blotting

Lysates for western blots were prepared at ZT0.5–1.5 from dissected adult brains in 50 µl Laemmli buffer as reported in *Parinejad et al., 2016*. 10 brains were used per lysate and incubated at 90°C for 5 min. 15 µl of each sample was loaded per well, run on 15% SDS-PAGE gels, blotted to nitrocellulose membrane, and probed with rabbit anti-AANAT1 (1:2500) or anti-Ebony (1:3000; Sean Carroll, University of Wisconsin-Madison), and mouse anti-actin (1:3000; Sigma #A4700). HRP-conjugated secondary antibodies anti-rabbit (1:3000; Bio-Rad) and anti-mouse (1:3000; Promega #W4021) were used for detection with chemiluminescence (HyGLO Chemiluminescent HRP Antibody Detection Reagent, Denville Scientific). Mean signal intensity for AANAT1 or Ebony was quantified using Fiji and normalized to actin. We used three separate lysates for each genotype to analyze western blots. For sleep experiments, female brains were used for lysate preparation.

## HPLC-MS

To prepare samples for HPLC-MS, the brains of twenty female flies (1–2 weeks old) for each genotype were dissected into ice-cold PBS between ZT0.5 and 3.5. We dissected brain tissue to avoid cuticle contamination because serotonin and dopamine are intermediates in the sclerotization of *Drosophila* cuticle. Dissected brains were centrifuged, the PBS was removed, and samples were quickly homogenized with a motorized pestle into an aqueous solution of formic acid (0.1%). After centrifugation, the supernatant was collected and stored at −80°C. Preliminary analytical conditions were developed using reference standards in a solution containing either serotonin, dopamine, or octopamine. With LC-MS/MS (Thermo-Scientific Quantiva Triple Quadrupole Mass Spectrometer (QQQ)), the absolute values for each analyte were measured in picograms (pg) per brain, through the addition of deuterated reference standards to sample extracts. All samples within an experiment were treated identically, and in parallel wherever possible.

## Monitoring and measurement of sleep in *Drosophila*

Prior to experimentation, flies were kept on standard food in constant conditions (a 12 hr light/dark cycle, and 25°C). At least 5 days after eclosion, mated adult females were loaded into glass tubes with 5% sucrose/2% agar food for behavioral recordings. The *Drosophila* Activity Monitoring (DAM) system (Trikinetics, Waltham, MA) was used to quantify infrared beam breaks representing locomotor activity. Files were processed with PySolo (*Gilestro and Cirelli, 2009*) in 1 min bins, with sleep defined as five consecutive minutes without activity, as done previously (*Hendricks et al., 2000*). In SD experiments, flies were placed in DAM monitors on a vortexer that was mechanically shaken a random 2 of every 20 s over the course of the 12 hr of the dark period (ZT12-24). Recovery sleep was determined, per fly, as the difference between sleep amount in the period following deprivation and sleep amount in the same time period on the preceding baseline day in unperturbed conditions. Activity Index refers to the average number of beam crossings within an active bout.

### Time-course of AANAT1 expression in astrocytes

AANAT1 levels in astrocytes were quantified at 3 hr intervals between ZT12-24 with IHC, where AANAT1 fluorescence intensity in astrocyte cell bodies was measured and normalized to nuRFP intensity from 2 copies of a UAS-nuRFP transgene reporter driven by Alrm-Gal4. At each time-point, 10 astrocytes in the antennal lobe region were measured from each of three brains.

## Acknowledgements

We thank Dr. Marc Freeman (Oregon Health and Science University) and Dr. Sean Carroll (University of Wisconsin-Madison) for reagents, to members of the van Meyel lab for helpful critique, and especially to Drs. Emilie Peco, Tiago Ferreira, Yimiao Ou, and Renu Heir for advice and assistance. HPLC-MS analysis was performed at the Proteomics Platform of the Research Institute of McGill University Health Centre. YL is supported in part by funding from the National Institutes of Health (NIH) grant DK120757, and AS is an Investigator of the Howard Hughes Medical Institute. This work was supported by grants to DvM from the Canadian Institutes of Health Research (CIHR, FRN159802), the Natural Sciences and Engineering Research Council of Canada (RGPIN05142), and the Canada Foundation for Innovation.

## Additional information

### Competing interests

Amita Sehgal: Reviewing editor, *eLife*. The other authors declare that no competing interests exist.

### Funding

| Funder | Grant reference number | Author |
| --- | --- | --- |
| Natural Sciences and Engineering Research Council of Canada | RGPIN-2017-05142 | Donald J van Meyel |
| Canadian Institutes of Health Research | MOP-137034 | Donald J van Meyel |
| National Institutes of Health | DK120757 | Amita Sehgal |

The funders had no role in study design, data collection and interpretation, or the decision to submit the work for publication.

### Author contributions

Sejal Davla, Conceptualization, Data curation, Formal analysis, Supervision, Funding acquisition, Validation, Investigation, Visualization, Methodology, Writing - original draft, Project administration, Writing - review and editing; Gregory Artiushin, Conceptualization, Data curation, Formal analysis, Validation, Investigation, Visualization, Methodology, Writing - original draft, Writing - review and editing; Yongjun Li, Data curation, Formal analysis, Validation, Investigation, Visualization, Methodology, Writing - review and editing; Daryan Chitsaz, Data curation, Formal analysis, Validation, Investigation, Visualization, Methodology, Writing - original draft, Writing - review and editing; Sally Li, Data curation, Formal analysis, Validation, Investigation, Methodology, Writing - review and editing; Amita Sehgal, Donald J van Meyel, Conceptualization, Supervision, Funding acquisition, Validation, Investigation, Visualization, Methodology, Writing - original draft, Project administration, Writing - review and editing

### Author ORCIDs

Sejal Davla https://orcid.org/0000-0003-3201-1970
Amita Sehgal https://orcid.org/0000-0001-7354-9641
Donald J van Meyel https://orcid.org/0000-0002-6075-8599

Decision letter and Author response
Decision letter https://doi.org/10.7554/eLife.53994.sa1
Author response https://doi.org/10.7554/eLife.53994.sa2

## Additional files

### Supplementary files
• Transparent reporting form

### Data availability
All data generated or analysed during this study are included in the manuscript and supporting files.

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
