## [Decision Letter]

**Acceptance summary:**

This is a very interesting study that adds to the growing body of work supporting a role for glial cells in sleep homeostasis. There are only a handful of studies in mammals and flies on this topic, so this study is timely and impactful. The demonstration of an astrocyte-specific role for AANT1 is backed up by a striking sleep phenotype, and the study benefits from the arsenal of tools available in this model.

**Decision letter after peer review:**

Thank you for submitting your article "AANAT1 functions in astrocytes to regulate sleep homeostasis" for consideration by *eLife*. Your article has been reviewed by three peer reviewers, and the evaluation has been overseen by a K VijayRaghavan as the Senior Editor and Reviewing Editor. The following individual involved in review of your submission has agreed to reveal their identity: Marcos G Frank (Reviewer #1).

The reviewers have discussed the reviews with one another and the Reviewing Editor has drafted this decision to help you prepare a revised submission.

Summary:

Using the Gal4-UAS system to knock-down AANAT1 in specific cell types, immuno-localization, and sleep monitoring, the authors show that this enzyme involved in monoamine degradation is expressed in a subpopulation of neurons and astrocytes, and is required in astrocytes to regulate monoamine and sleep homeostasis. Baseline sleep and baseline monoamine levels do not appear to be affected by the loss of AANAT1. Little is known regarding how sleep regulation, glia, and monoamine signaling are connected and thus these data are timely. Overall this is a very interesting study that adds to the growing body of work supporting a role for glial cells in sleep homeostasis. There are only a handful of studies in mammals and flies on this topic, so this study is timely and impactful. The demonstration of an astrocyte-specific role for AANT1 is backed up by a striking sleep phenotype, and the study benefits from the arsenal of tools available in this model.

The work is generally well-conducted, but there are a few weak points that need to be addressed to consolidate and enhance the conclusions.

Essential revisions:

1) The authors show that the number of astrocytes is not modified upon AANAT1 knockdown, however a knockdown induced specifically at the adult stage would confirm that the effect observed is not the consequence of developmental defects, or alternatively, attenuated by compensatory effects induced by the constitutive inhibition. This could be achieved using a time-controlled version of the Gal4-UAS system.

2) The HPLC data shown in Figure 4J show values that are much smaller than the ones shown in Figure 2I. The authors should explain why such a difference is observed. This may indicate that the knockdown of AANAT1 in astrocytes is significantly reducing monoamines levels in the brain, a finding that would modify the interpretation of the data. This conclusion would require to evaluate monoamines in the genetic controls, currently not shown in this experiment. This should be clarified.

3) The authors do not detect a variation of global AANAT1 expression levels upon sleep deprivation, however, they observe a change in expression across time of day. These latter data (Figure 4K) are based on the quantification of AANAT1 immunofluorescence in astrocytes, however it would be important to know in which brain region/structure the signal was assessed and this is not specified. In addition, these data cannot be compared to the sleep deprivation data obtained in whole brain extract (Figure 2J) that include expression in neurons. This information is important to assess whether AANAT1 expression is linked to sleep homeostasis. Secondly, there is no time of day information regarding the immuno-localizations shown in Figure 1. Given that the expression is not uniform in astrocytes and apparently absent from some sleep/wake regulatory structures (Figure 1), it seems highly relevant to assess whether or not this conclusion holds true at different times of day.

4) It is unclear how the metabolism of monoamines involving Ebony, Black and Tan is related to their inactivation by AANAT1 in terms of sleep regulation. This is not clearly shown and discussed in the paper.

5) The authors do not mention possible circadian changes of 5-HT, DA and AANAT1 levels in the brain. The obtained results are not always statistically significant, for example in comparing the experimental RNAi line with control.

6) We suggest that the authors consider AANAT1 over-expression in glial cells on sleep and 5-HT and DA levels before and after sleep deprivation to strengthen their conclusion.

---

## [Author Response]

Essential revisions:1) The authors show that the number of astrocytes is not modified upon AANAT1 knockdown, however a knockdown induced specifically at the adult stage would confirm that the effect observed is not the consequence of developmental defects, or alternatively, attenuated by compensatory effects induced by the constitutive inhibition. This could be achieved using a time-controlled version of the Gal4-UAS system.

To address this, we used the TARGET system to knock down AANAT1 with Eaat1-Gal4. In the brain, Eaat1-Gal4 is specific for astrocytes (which express AANAT1) and cortex glia (which do not). The TARGET system allowed us to use a temperature shift to genetically restrict the GAL4-UAS system to the adult stage only. In this experiment, AANAT1 knockdown in adult astrocytes increased recovery sleep compared to the UAS control but not compared to the Gal4 control. We have included these results in the revised manuscript (Figure 4—figure supplement 1J,K).

We also considered the drug-regulated GeneSwitch system to temporally restrict the GAL4-UAS system to adult glial cells. However, the only existing tool for this is a Repo-GeneSwitch line, and it is not appropriate for this application because the Repo promoter is expressed poorly in astrocytes in adults. We tested it anyway and, predictably, we saw no effect on sleep recovery with Repo-GeneSwitch. These data are not included in the revised manuscript.

As an alternative approach to address the reviewers’ question, we did a time-course of AANAT1 expression through pupal stages, using immunochemistry (Figure 4—figure supplement 1G-I’). Our results show that AANAT1 is expressed weakly in only a few astrocytes at 48h after puparium formation (APF), then more strongly in most but not all astrocytes at 72h and 96h APF.

In summary, the results of the TARGET experiment provide some support for the idea that AANAT1 functions in adult astrocytes to limit recovery sleep, but they do not cleanly eliminate the possibility of a developmental role for AANAT1. However, AANAT1 expression in astrocytes appears gradually during pupal development, and is sustained at high levels in all astrocytes in adults (Figure 1F). Loss of AANAT1 from astrocytes preferentially affects sleep homeostasis, but it does not affect general sleep patterns or baseline sleep levels. Since loss of AANAT1 does not affect astrocyte numbers or their terminal differentiation, we think it unlikely to play a developmental role and favor the idea that AANAT1 functions in mature astrocytes to limit recovery sleep in adults. We added this to the Results and Discussion.

2) The HPLC data shown in Figure 4J show values that are much smaller than the ones shown in Figure 2I. The authors should explain why such a difference is observed. This may indicate that the knockdown of AANAT1 in astrocytes is significantly reducing monoamines levels in the brain, a finding that would modify the interpretation of the data. This conclusion would require to evaluate monoamines in the genetic controls, currently not shown in this experiment. This should be clarified.

Average levels of monoamines in the lysates of pooled brains are indeed lower in 4J controls than 2I controls, though there is some overlap in the distribution of data points. As the lower levels in 4J were noted not only in experimental animals, but also in Gal4 and UAS controls (in fact, several UAS control samples were below the limit of detection and so were excluded from analysis), they do not result from knockdown of AANAT1 in astrocytes. This has been clarified in the figure legend for Figure 4J. Most likely, the variations result from different genetic backgrounds, despite outcrossing to iso31. Efficiency of extraction/detection could also contribute, as the experiments in 2I and 4J were done independently of one another, separated in time by several months. Note that all samples within an experiment were treated identically, and in parallel wherever possible. This point has been clarified in the Materials and methods.

3) The authors do not detect a variation of global AANAT1 expression levels upon sleep deprivation, however, they observe a change in expression across time of day. These latter data (Figure 4K) are based on the quantification of AANAT1 immunofluorescence in astrocytes, however it would be important to know in which brain region/structure the signal was assessed and this is not specified. In addition, these data cannot be compared to the sleep deprivation data obtained in whole brain extract (Figure 2J) that include expression in neurons. This information is important to assess whether AANAT1 expression is linked to sleep homeostasis. Secondly, there is no time of day information regarding the immuno-localizations shown in Figure 1. Given that the expression is not uniform in astrocytes and apparently absent from some sleep/wake regulatory structures (Figure 1), it seems highly relevant to assess whether or not this conclusion holds true at different times of day.

We thank the reviewers for pointing to these omissions. We added these important details (underlined above) to the revised manuscript.

Firstly, the signal in Figure 4K was assessed in astrocyte cell bodies in the anterior-dorsal antennal lobe and in the central complex nearby. We chose these brain regions/structures because they routinely showed excellent immunochemical signal and good cellular resolution. We agree that the data in Figure 4K (immunochemistry ZT12-ZT24, with no sleep deprivation) cannot be compared to the data in Figure 2J (Western blot, which includes neurons in lysates, with sleep deprivation), and did not intend to do so. In the revised manuscript, we simply “speculate that the loss of AANAT1 might have profound influence on sleep homeostasis near ZT15, when AANAT1 levels in astrocytes are usually highest, or during daytime when the increased recovery sleep occurs.”

Secondly, regarding the time of day information for Figure 1 and its supplementary figure – these levels were assessed during the day between ZT3-ZT9. We have never seen obvious changes of AANAT1 levels in astrocytes in the light phase. In contrast, in the dark phase we did see obvious and widespread decrease of AANAT1 levels in astrocytes – and our AANAT1 quantification of astrocytes in the antennal lobe and central complex confirms this (Figure 4K). Known sleep/wake regulatory structures (mushroom body, fan-shaped body, and ellipsoid body) were also examined where, in the dark phase as in the light phase, AANAT1 expression was largely absent from the neuropils of the mushroom body and fan-shaped body, and AANAT1 expression in the neuropil of the ellipsoid body was neuronal. A statement regarding dark phase expression in these areas has now been added to the revised manuscript.

4) It is unclear how the metabolism of monoamines involving Ebony, Black and Tan is related to their inactivation by AANAT1 in terms of sleep regulation. This is not clearly shown and discussed in the paper.

In the central brain, Ebony is known to be expressed in astrocytes, and could provide an alternative mechanism for monoamine catabolism (via NBAD). As outlined in our manuscript, we tested for Ebony levels with Western blot and found they are not altered to compensate for loss of AANAT1 from astrocytes, nor are Ebony levels affected by sleep deprivation (Figure 2—figure supplement 1A, C). Therefore, the involvement of Ebony with AANAT1 in sleep homeostasis, if any, was not further addressed. To address the reviewers’ point about Black and Tan, we looked at single-cell RNA sequencing (scRNAseq) data from the lab of Dr. Stein Aerts and their web-based visualization tool known as SCope. In scRNAseq data from 57,000 adult brain cells, there was no evidence for Black or Tan expression in astrocytes and so we did not consider their involvement further.

5) The authors do not mention possible circadian changes of 5-HT, DA and AANAT1 levels in the brain. The obtained results are not always statistically significant, for example in comparing the experimental RNAi line with control.

Related to circadian changes of 5-HT and DA – Since we did not find changes in baseline sleep rhythms or parameters, we did not make a priority of quantifying the monoamines 5-HT and DA with HPLC-MS for a full circadian cycle, a costly effort requiring substantial time and human resources. We have not mentioned it because we believe this to be beyond the scope of this manuscript, and that is why we focused on changes in these monoamines upon sleep deprivation instead.

Related to circadian changes of AANAT1 – In Figure 4K, we showed time-of-day changes in AANAT1 levels in astrocytes, as described above. For total AANAT1 levels in the brain (as perhaps suggested in the reviewers’ comment), we would need to do Western blots but, as noted earlier by the reviewers, this would include neurons and would not be directly related to our understanding of AANAT1 in astrocytes.

Related to statistical significance- we believe this comment refers to Figure 4J, where astrocyte-selective AANAT1 knockdown led to increased levels of brain serotonin and dopamine after sleep deprivation in most samples compared to controls. However, as we noted, this did not reach statistical significance. We raised two possibilities to explain this. First, there is one outlier in each of the control groups. Second, AANAT1 knockdown in astrocytes is likely to affect the level of only a portion of serotonin and dopamine in the brain. Although perhaps not enough to establish a robust statistical difference between samples taken from whole brain lysates, we think it is an important portion with respect to sleep homeostasis, acting in a brain region-specific or circuit-specific manner.

6) We suggest that the authors consider AANAT1 over-expression in glial cells on sleep and 5-HT and DA levels before and after sleep deprivation to strengthen their conclusion.

We did the suggested experiment and found that AANAT1 overexpression in astrocytes increases recovery sleep (Figure 4—figure supplement 1), which phenocopies the effect of AANAT1 loss-of-function. This data underscores the importance of regulated astrocytic AANAT1 levels in sleep homeostasis and raises the possibility that AANAT1 overexpression could have dominant-negative effect.